

# Elevational patterns of the percentages of plant genera with tropical and temperate affinities in Nepal

Yunyun Lai and Jianmeng Feng

School of Life Science and Agronomy, Dali University, Dali, Yunnan, China

## ABSTRACT

**Background:** Geographical patterns of species diversity are one of the key topics in biogeography and ecology. The effects of biogeographical affinities on the elevational patterns of species diversity have attracted much attention recently, but the factors driving elevational patterns of the percentages of plants with tropical and temperate biogeographical affinities have not been adequately explored.

**Methods:** We first used univariate least squares regressions to evaluate the effects of each predictor on the elevational patterns of the percentages of plant genera with tropical and temperate affinities in Nepal. Then, the lowest corrected Akaike information criterion value was used to find the best-fit models for all possible combinations of the aforementioned predictors. We also conducted partial regression analysis to investigate the relative influences of each predictor in the best-fit model of the percentages of plant genera with tropical and temperate affinities.

**Results:** With the increase of elevation, the percentage of plant genera with tropical affinity significantly decreased, while that of plant genera with temperate affinity increased. The strongest predictor of the percentages of plant genera with tropical affinity in the examined area was the minimum temperature of the coldest month. For the elevational patterns of the percentages of plant genera with temperate affinity, the strongest predictor was the maximum temperature of the warmest month. Compared with mid-domain effects (MDE), climatic factors explained much more of the elevational variation of the percentages of plant genera with tropical and temperate affinities.

**Discussion:** The elevational patterns of the percentages of plant genera with tropical affinities and the factors driving them supported the revision of the freezing-tolerance hypothesis. That is, freezing may filter out plant genera with tropical affinity, resulting in the decrease of their percentages, with winter coldness playing a predominant role. Winter coldness may not only exert filtering effects on plant genera with tropical affinity, but may also regulate the interactions between plant genera with tropical and temperate affinities. The elevational patterns of tropical and temperate plant diversities, and those of their percentages, might be controlled by different factors or mechanisms. Freezing-tolerance and the interactions between plant genera with tropical and temperate affinities regulated by climatic factors played stronger roles than MDE in shaping the elevational patterns of the percentages of plant genera with tropical and temperate affinities in Nepal.

Corresponding author
Jianmeng Feng, fjm@pku.org.cn

## INTRODUCTION

To a certain extent, biogeographical affinities of organisms reflect their evolutionary history, eco-physiological traits, and adaptation to environmental factors. These affinities may therefore partly determine the distribution patterns of organisms (*Latham & Ricklefs, 1993a*, *1993b*; *Wiens & Donoghue, 2004*; *Wang et al., 2011*; *Feng et al., 2016*). For that reason, elevational patterns of diversity and the percentages of plants with one specific biogeographical affinity may differ from those of plants with other biogeographical affinities. However, as far as we know, only the effects of biogeographical affinities on the elevational patterns of species diversity have attracted attention. Relevant studies on this subject showed that although diversity of plants with tropical affinities increased and then decreased with the increase of elevation, showing hump-shaped patterns, it peaked at lower elevations than temperate species diversity (*Oommen & Shanker, 2005*; *Wang, Tang & Fang, 2007*; *Feng et al., 2016*). This difference may be at least partly due to different biogeographical affinities and related eco-physiological traits, which may modify the influence of water and energy-related variables on the elevational patterns of species diversity (*Oommen & Shanker, 2005*; *Wang, Tang & Fang, 2007*; *Feng et al., 2016*). We noted that, although several studies showed that the percentages of plant genera with tropical and temperate affinities linearly decreased and increased with the increase of elevation, respectively (*Wang, Tang & Fang, 2007*; *Feng & Xu, 2008a*, *2008b*; *Xu & Feng, 2010*), the associated controlling factors or underlying mechanisms have been poorly statistically tested. For example, as elevation has long been considered as a proxy of energy gradient, the decreasing trends of the percentages of warmth preferring plants with tropical affinity may be highly linked with energy factors associated with elevation. However, as far as we know, this has not been statistically tested.

The hypothesis of mid-domain effects (MDE) suggests that hump-shaped patterns of diversity along elevational gradients are caused by the increasing overlap of species ranges toward the center of the elevational gradients (*Colwell & Lees, 2000*; *Colwell, Rahbek & Gotelli, 2004*). Recently, this hypothesis has been pervasively used to explain hump-shaped diversity patterns, although its influence and generality remain debatable (*Colwell, Rahbek & Gotelli, 2005*; *Hawkins, Diniz-Filho & Weis, 2005*; *Herzog, Kessler & Bach, 2005*; *Kluge, Kessler & Dunn, 2006*). Previous studies in Nepal observed that the relative influences of MDE and climatic factors on species diversity might vary with the biogeographical affinities of plants (*Li & Feng, 2015*). However, no study conducted so far has investigated their relative roles on the elevational patterns of the percentages of plants with tropical and temperate affinities.

The freezing-tolerance hypothesis states that winter coldness may primarily determine spatial patterns of species diversity because most clades evolved in tropical-like climate or have tropical affinities and hence could hardly disperse into cold, temperate regions; compared with that of temperate affinities, the richness of the species with tropical

affinities may be more strongly influenced by winter coldness (*Latham & Ricklefs, 1993a*, *1993b*; *Wiens & Donoghue, 2004*; *Wang et al., 2011*). Based on this hypothesis, with increasing elevation, which is a negative proxy of energy, tropical plant diversity may decrease, and winter coldness may be a driver of it. However, not all relevant studies have supported this hypothesis. For example, in Nepal, *Li & Feng (2015)* found hump-shaped patterns of tropical plant diversity on an elevational gradient, and this effect was mainly controlled by water availability. In addition, *Wang, Tang & Fang (2007)* also observed hump-shaped elevational patterns of tropical plant diversity in Gaoligong Mountains, Southeast Tibet, and these were mainly controlled by area and MDE. In the present study, we proposed and corroborated a revised version of the freezing-tolerance hypothesis, which asserts that freezing may filter out plants with tropical affinity, resulting in a decrease in their percentage in regional floras along elevational gradients and that winter coldness may play predominant roles in this pattern. Given this revised version of the freezing-tolerance hypothesis, the percentage of plant genera with tropical affinity may decrease with the increase of elevation and freezing effects.

In the present study, we investigated the elevational patterns of the percentages of plant genera with tropical and temperate affinities in Nepal and the associated controlling factors. We aimed to obtain the relative roles of MDE and climatic factors on the elevational patterns of tropical and temperate plant percentages, and to test the following hypothesis: the elevational patterns shown by the percentages of plant genera with tropical affinity may support the predictions of the revised version of the freezing-tolerance hypothesis.

# MATERIALS AND METHODS

## Study area

Our study area covered 157,130 km$^2$ in the region between 300 and 5,700 m a.s.l., located on the southern slopes of the Himalayas in Nepal (Fig. 1). In the south to north direction, the elevation increases from roughly 300 m a.s.l. to more than 5,700 m a.s.l., and environmental energy decreases by 0.55 °C per 100 m of elevation from approximately 19.4 °C at 1,000 m a.s.l. (*Feng et al., 2016*). Annual precipitation at 1,600–1,700 m a.s.l. is high, and it decreases with increasing elevation; all seasonal variables (i.e., seasonal temperature, seasonal precipitation, and annual temperature range) show open-upward-parabola patterns, that is, decreasing and then increasing along the elevational gradient from 300 to 5,700 m a.s.l (*Feng et al., 2016*). In South Nepal, regions below 2,000 m a.s.l. are mainly characterized by subtropical climate; in Central Nepal, regions between 2,000 and 4,000 m a.s.l. are primarily dominated by temperate or warm temperate climate; and in North Nepal regions above 4,000 m a.s.l. have an alpine climate (*Dobremez, 1976*; *Bhattarai & Vetaas, 2003*). Consistently, with the increase in latitude and elevation, vegetation changes from subtropical to warm temperate, temperate, and eventually alpine (*Dobremez, 1976*; *Bhattarai & Vetaas, 2003*).

## Plant data

Species identity, genus, and family, as well as the minimum and maximum elevation of each species, were obtained from the online version of the Annotated Checklist of the

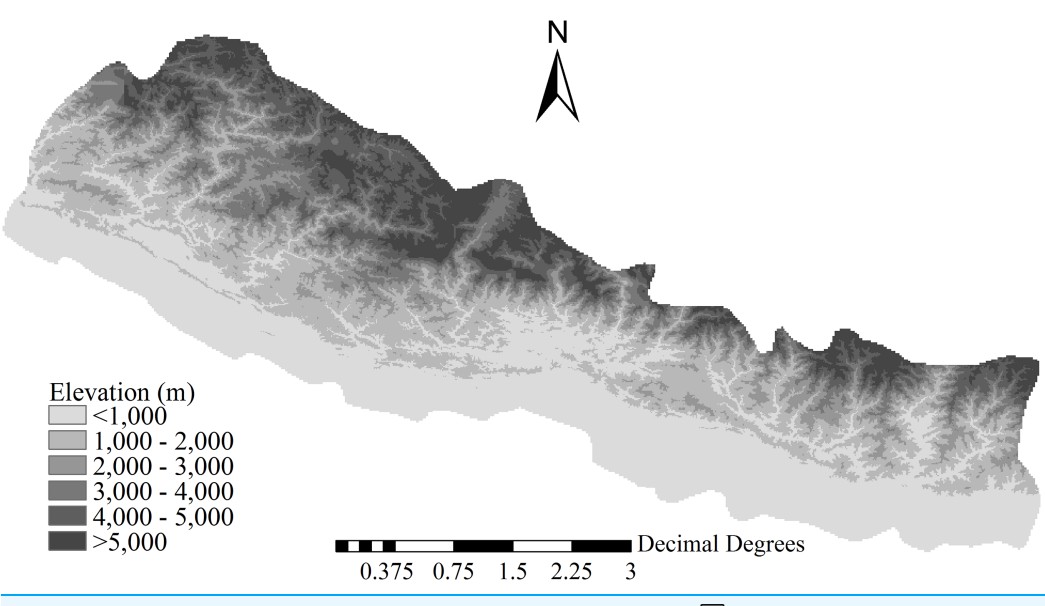

Elevation (m)
- <1,000
- 1,000 - 2,000
- 2,000 - 3,000
- 3,000 - 4,000
- 4,000 - 5,000
- >5,000

0.375  0.75  1.5  2.25  3    Decimal Degrees

**Figure 1 Location and topography of the study area.**

Flowering Plants of Nepal (*Press, Shrestha & Sutton, 2016*). In the study, all varieties and subspecies were considered as separate species. To eliminate or reduce the effects of human disturbance at low elevations and low sampling effort at high elevation, we only considered plants sampled between 300 and 5,700 m a.s.l., that is, 4,759 species, belonging to 1,382 genera and 218 families.

## Biogeographical affinities

We consulted *Wu's (1991)* dictionary to determine the biogeographical affinities of seed plant genera in Nepal. In this classification system, the biogeographical affinities of genera are primarily defined or determined based on their biogeographical history, fossil records, and modern distribution centers, in particular. For example, if the distribution center of a given genus is in a tropical region, such as a tropical rainforest climate zone, tropical monsoon climate zone, tropical wet and dry or savanna climate zone, or hot desert, this genus is designated a tropical genus or genus with tropical affinities. As the study area was adjacent to China, and the two regions more likely belong to the Eurasian Plate than to the Indian Plate biogeographical zone, this system of biogeographical affinities might be applied to Nepal, as suggested in a previous study (*Li & Feng, 2015*). In addition, we consulted the system of biogeographical affinities created by *Wielgorskaya (1995)*. We found that more than 96.6% of the biogeographical affinities (tropical or temperate affinity) of seed plant genera in China defined by Wu's system (1991) were consistent to those determined by the system developed by *Wielgorskaya (1995)*. For 43 genera whose biogeographical affinities are not included in Wu's biogeographical affinities, we used the system of biogeographical affinities created by *Wielgorskaya (1995)* to determine their biogeographical affinities. Our approach was thus similar to that used in a variety of studies (*Qian, 1999*, *2001*; *Wang et al., 2011*; *Li & Feng, 2015*) that drew robust conclusions. Overall, we obtained 710 plant genera

with tropical affinities and 516 plant genera with temperate affinity between 300 and 5,700 m a.s.l. in Nepal. As the number of plant genera with tropical and temperate affinities composed 88.7% of the total number of genera, we only considered genera with tropical and temperate biogeographical affinities in the present study.

## Interpolation of plant occurrence

The occurrence of each genus present between 300 and 5,700 m a.s.l. was interpolated based on the recorded elevational range of each genus to obtain the number of tropical, temperate and total genera in each 100 m elevation band (e.g., 300–400 m a.s.l., 400–500 m a.s.l.) for a total of 54 elevational bands. This method, which has been frequently used in previous studies (*Rahbek, 1997*; *Grytnes & Vetaas, 2002*), assumed that the genera were distributed continuously between their minimum and maximum elevations (i.e., present in every 100 m vertical band). Then, we estimated the percentage of plant genera with tropical and temperate affinities in each elevational band. The percentage of the plant genera with tropical and temperate affinities was calculated as the ratio between the numbers of plant genera with tropical and temperate affinities and the total number of genera in each elevational band, respectively, and each ratio was multiplied by 100. The percentages were log-transformed to improve data normality.

## Predictors on elevation

The three categories of climatic variables used in the present study were as follows: (i) energy availability, represented by the mean annual temperature (MAT, °C), mean temperature of the coldest quarter (MTCQ, °C), mean temperature of the warmest quarter (MTWQ, °C), mean temperature of the driest quarter (MTDRYQ, °C), maximum temperature of the warmest month (MTWM, °C), and minimum temperature of the coldest month (MTCM, °C) (Fig. 2); (ii) water availability, comprising mean annual precipitation (MAP, mm), precipitation in the driest quarter (PDQ, mm), precipitation in the warmest quarter (PWARMQ, mm), and precipitation in the wettest quarter (PWETQ, mm) (Fig. 2); (iii) climatic seasonality, indicated by the annual range of temperature (ART, °C; MTWM—MTCM), temperature seasonality (TSN, °C), and precipitation seasonality (PSN, mm; coefficient of variation of the mean monthly precipitation) (Fig. 2). In addition to the climatic factors, we also included the plant percentages generated by MDE in the present study which were tropical plant percentages generated by MDE (TRPMDE) and temperate plant percentages generated by MDE (TEPMDE) (Fig. 3). All climatic data (1970–2000) were downloaded from WorldClim at a resolution of 30 arc-second (ca. one km at the equator) (*Hijmans et al., 2005*). Although this dataset was derived through spatial interpolation, it could reliably reflect elevational variation of the climatic factors, as confirmed by *Qian (2014)* and in studies on elevational patterns of plant diversity in Nepal (*Li & Feng, 2015*; *Feng et al., 2016*). We used the Mid-Domain Null Program (*McCain, 2004*) to predict null percentages of plant genera with tropical and temperate affinities, respectively. Firstly, we simulated genus richness over the elevational gradient for total plant genera with tropical and temperate affinities separately. To eliminate bias caused by the differences between theoretical frequency distributions of

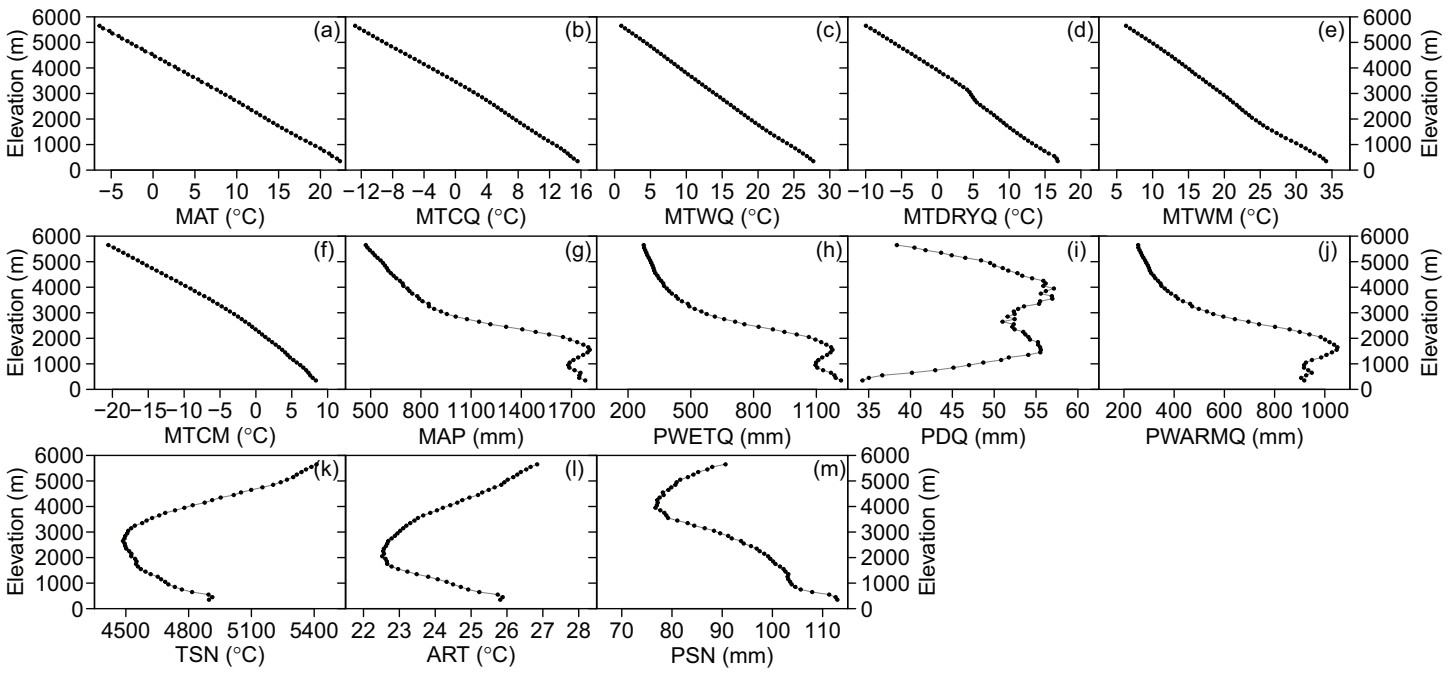

**Figure 2 Elevational patterns of climatic factors in Nepal.** (A) MAT, mean annual temperature; (B) MTCQ, mean temperature of the coldest quarter; (C) MTWQ, mean temperature of the warmest quarter; (D) MTDRYQ, mean temperature of the driest quarter; (E) MTWM, maximum temperature of the warmest month; (F) MTCM, minimum temperature of the coldest month; (G) MAP, mean annual precipitation; (H) PWETQ, precipitation in the wettest quarter; (I) PDQ, precipitation in the driest quarter; (J) PWARMQ, precipitation in the warmest quarter; (K) TSN, temperature seasonality; (L) ART, annual range of temperature; (M) PSN, precipitation seasonality.

range sizes and the observed ones, we used observed range sizes without replacement and randomly chosen range midpoints to produce ranges within the domain limits. Then, we estimated null elevational percentages of plant genera with tropical and temperate affinities as predicted by MDE separately (Fig. 3). The percentages of plant genera with tropical and temperate affinity as predicted by MDE were calculated as the ratio between the numbers of plant genera with tropical and temperate affinities predicted by MDE and the total number of genera predicted by MDE in each elevational band, respectively, and each ratio was multiplied by 100.

## Statistical methods

All the statistical analyses were performed using SAM v4.0 (*Rangel, Diniz-Filho & Bini, 2010*). We first used univariate least squares regression (LSR) to evaluate the effects of the three categories of climatic factors on the elevational patterns of the percentages of plant genera with tropical and temperate affinities, respectively. We generated regression models of all possible combinations of the aforementioned predictors, and then we found the best-fit models on the basis of the lowest possible corrected Akaike information criterion (AIC$_c$) values, which deals with the trade-off between the goodness of fit of the model and its simplicity (*Burnham & Anderson, 2002*). The following methodologies were used to select the candidate predictors: (1) only the best predictor in each predictor group in univariate LSR models could be a candidate, thereby reducing or avoiding

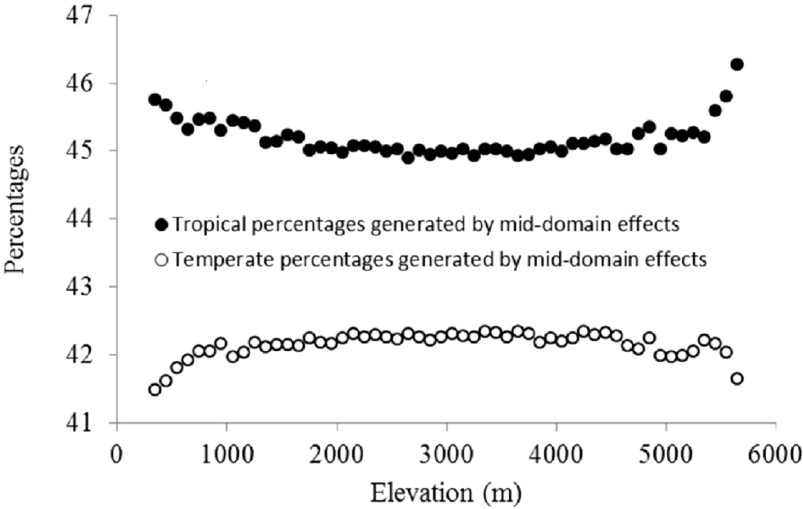

**Figure 3 Elevational patterns of the percentages of tropical and temperate genera generated by MDE.**

multicollinearity among predictors in the same predictor group; (2) only the predictor showing significant effects in univariate LSR models was allowed to enter the model; (3) the predictor showing the highest explanatory power in all univariate LSR models was included in all of the best-fit models. Following *Burnham & Anderson (2002)*, the model with the lowest $AIC_c$ was defined as the best model, and if a model with $\Delta AIC_c < 2$ in comparison with the best-fit model, it was considered equally good fit to the best-fit model.

Residual dependence resulting from spatial autocorrelation in ecological data may cause inflated type I errors which may inflate significance level of statistical tests (*Lennon, 2000*). Therefore, spatial linear autoregressive models (SLMs) were frequently used to account for the inflated significance level. However, our major topic was not the significance levels but rather the explanatory powers of the predictors. In addition, the influence of predictors derived from SLMs may depend on the scales of spatial structures; the influences of predictors on large or small-scale spatial structures may be underestimated or overestimated, respectively (*Diniz-Filho, Bini & Hawkins, 2003*). Therefore, SLMs were not used in the present study.

## RESULTS

The percentage of plant genera with tropical affinity along the elevational gradient significantly decreased from more than 80% to close to zero, while that of plant genera with temperate affinity increased with the increase of elevation (Fig. 4). The univariate regressions models showed that MTCM was the strongest predictor among the energy variables for the percentage of plant genera with tropical affinity (98.6% of the variation explained) along the elevational gradient; MAP (91.3%) and PSN (71.4%) were the strongest predictor in the category of water variables and climatic seasonality variables, respectively (Table 1). Therefore, MTCM, a factor representing winter freezing, was selected as the best predictor for the elevational patterns of the percentage of plant genera with tropical affinity. For the percentage of plant genera with temperate affinity,

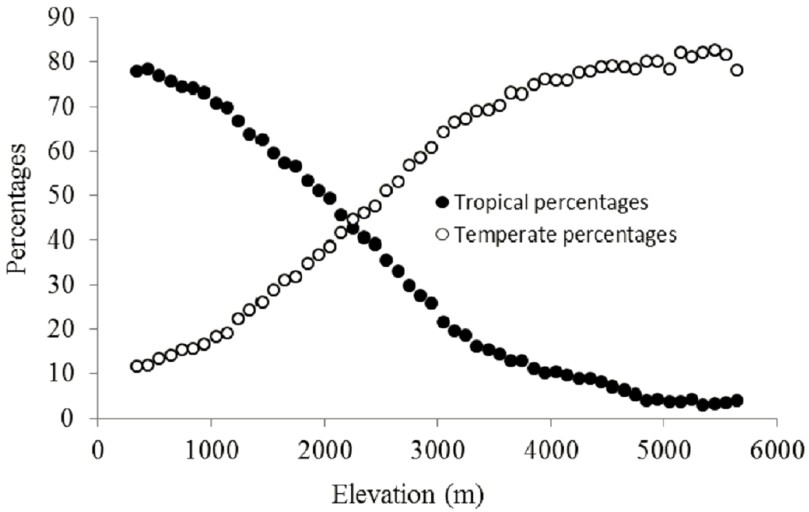

**Figure 4 Elevational patterns of the percentages of tropical and temperate genera in Nepal.**

MTWM was the strongest predictor among the energy variables (87.7% of the variation explained), while PWETQ (84.6%) and PSN (86.3%) were the strongest predictors in the categories of water variables and climatic seasonality variables, respectively (Table 1). The univariate regressions models showed that neither TRPMDE nor TEPMDE showed significant effects on the elevational variation of the percentage of plant genera with tropical affinity (Table 1). Therefore, no predictor from the category of the percentages generated by MDE was selected as candidate in the $AIC_c$ based best-fit models for the percentages of plant genera with tropical affinity. Neither TRPMDE nor TEPMDE showed high explanatory power on the elevational variation of the percentages of plant genera with temperate affinity (26.5% ($P < 0.01$) and 11.2% ($P < 0.05$), respectively) (Table 1), but TEPMDE from the category of the percentages generated by MDE was selected as a candidate in the $AIC_c$ based best-fit model for the percentage of plant genera with temperate affinity.

The $AIC_c$ based model selection showed that for the percentages of plant genera with tropical affinity there were two equally-good multiple regressions models (S1). The first comprised MTCM and MAP ($\Delta AIC_c = 0$), and 98.8% of the variation was explained (S1). The second comprised MTCM, MAP, and PSN ($\Delta AIC_c = 1.7$), and 99.8% of the variation was explained (S1). For the percentages of plant genera with temperate affinity the best multiple regressions model comprised MTWM, PSN, and TEPMDE, and 96.2% of the variation was explained (S1).

Partial regression analyses using variables from the first best models for the percentages of plant genera with tropical affinity showed that MTCM totally and independently explained 98.6% and 7.5% of the elevational variation of the percentages of plant genera with tropical affinity, respectively, while for MAP, the total and independent percentages explained were 91.3% and 0.2%, respectively (Fig. 5). The total and shared percentage explained were 98.8% and 91.1%, respectively (Fig. 5).

**Table 1 Explanatory power of predictors for the percentages of tropical and temperate plants on elevational patterns through univariate regressions models (%).**

| Predictors | | | The percentages of tropical plants | | The percentages of temperate plants | |
|---|---|---|---|---|---|---|
| | | | Explanatory power (%) | STD error | Explanatory power (%) | STD error |
| Climatic factors | Energy factor | MAT (°C) | 98.2*** | 0.114 | −85.0*** | 0.251 |
| | | MTCQ (°C) | 97.7*** | 0.164 | −82.7*** | 0.269 |
| | | MTWQ (°C) | 95.9*** | 0.219 | −87.3*** | 0.231 |
| | | MTDRYQ (°C) | 96.5*** | 0.201 | −83.1*** | 0.266 |
| | | MTWM (°C) | 94.8*** | 0.247 | −87.7*** | 0.219 |
| | | MTCM (°C) | 98.6*** | 0.131 | −78.1*** | 0.303 |
| | Water factors | MAP (mm) | 91.3*** | 0.324 | −82.1*** | 0.273 |
| | | PWETQ (mm) | 89.5*** | 0.349 | −84.6*** | 0.253 |
| | | PDQ (mm) | −0.019$^{ns}$ | 1.088 | 16.0** | 0.593 |
| | | PWARMQ (mm) | 90.2*** | 0.337 | −73.1*** | 0.336 |
| | Seasonal factors | TSN (°C) | −57.0*** | 0.707 | 9.8** | 0.615 |
| | | ART (°C) | −34.8*** | 0.870 | 0.50$^{ns}$ | 0.648 |
| | | PSN (mm) | 71.4*** | 0.585 | −86.3*** | 0.243 |
| PMDE | TRPMDE | | −1.8$^{ns}$ | 1.106 | 11.2** | 0.591 |
| | TEPMDE | | −0.3$^{ns}$ | 1.098 | 26.5*** | 0.542 |

Notes:
MAT, the mean annual temperature (°C); MTCQ, mean temperature of the coldest quarter (°C); MTWQ, mean temperature of the warmest quarter (°C); MTDRYQ, mean temperature of the driest quarter (°C); MTWM, maximum temperature of the warmest month (°C); MTCM, minimum temperature of the coldest month (°C); MAP, mean annual precipitation (mm); PWETQ, precipitation of the wettest quarter (mm); PDQ, precipitation in the driest quarter (mm); PWARMQ, precipitation in the warmest quarter (mm); TSN, temperature seasonality (°C); ART, annual range of temperature (°C); PSN, precipitation seasonality (mm); PMDE, percentages generated by MDE; TRPMDE, tropical percentages generated by MDE; TEPMDE, temperate percentages generated by MDE.
*** $P < 0.01$.
** $P < 0.05$.
ns $P > 0.05$.

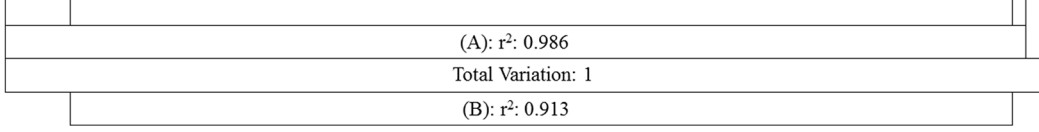

**Figure 5 Comparing the explanatory power of MTCM and MAP on the percentages of plant genera with tropical affinity by partial regression with the predictors included in the first best mode.** A Shows the effects of MTCM; B shows the effects of MAP. Total variance explained by {A} = 98.6%; Total variance explained by {B} = 91.3%; Total variance explained by {A+B} = 98.8%. [A.B] variance explained by {A} only = 7.5%; [A:B] Variance Sharely explained = 91.1%; [B.A] Variance explained by {B} only = 0.2%; [1−(A+B)] Unexplained variance = 1.2%.

Partial regression analyses using variables from the second best models for the percentages of plant genera with tropical affinity showed that MTCM totally and independently explained 98.6% and 6.9% of the elevational variation of the percentages of plant genera with tropical affinity, respectively (Fig. 6); MAP totally and independently explained 91.3% and 0.2% of the elevational variation, respectively; PSN totally and independently explained 71.4% and <0.1% of the elevational variation, respectively. Their total and shared percentage explained was 98.8% and 70.7%, respectively (Fig. 6).

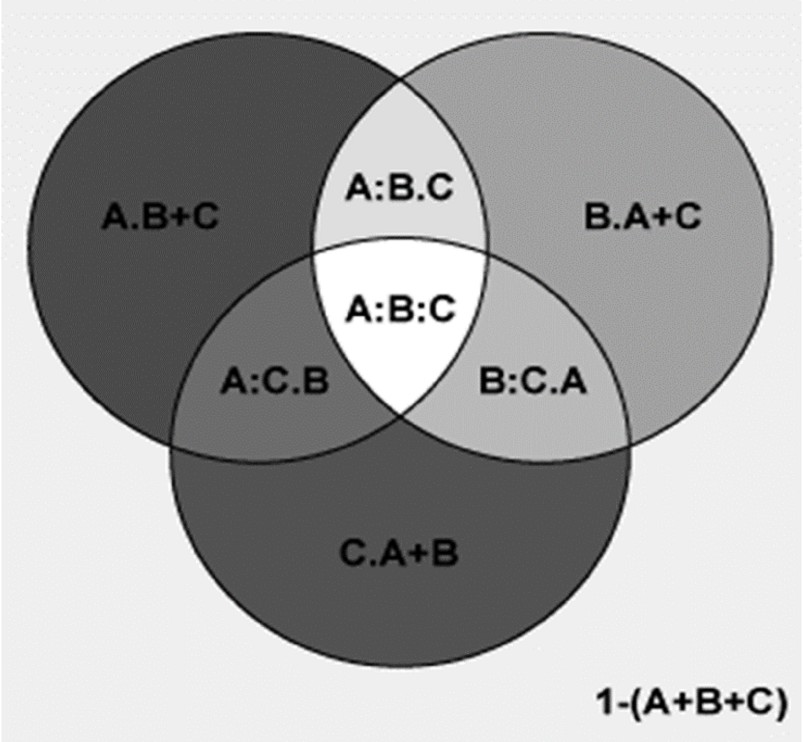

**Figure 6 Comparing the explanatory power of MTCM, MAP, and PSN on the percentages of plant genera with tropical affinity by partial regression with the predictors included in the second best model.** A shows the effects of MTCM; B shows the effects of MAP; C shows the effects of PSN. A = 0.986; 1−(A) = 0.014; A.B = 0.075; A:B = 0.911; A.C = 0.273; A:C = 0.713; A.B+C = 0.069; A:B+C = 0.918; B = 0.913; 1−(B) = 0.087; B.A = 0.002; B:A = 0.911; B.C = 0.206; B:C = 0.707; B.A+C = 0.002; B:A+C = 0.912; C = 0.714; 1−(C) = 0.286; C.A = <0.001; C:A = 0.713; C.B = 0.006; C:B = 0.707; C.A+B = <0.001; C:A+B = 0.714; A+B = 0.988; 1−(A+B) = 0.012; A+B.C = 0.274; A+B:C = 0.714; A+C = 0.987; 1−(A+C) = 0.013; A+C.B = 0.075; A+C:B = 0.912; B+C = 0.92; 1−(B+C) = 0.08; B+C.A = 0.002; B+C:A = 0.918; A+B+C = 0.988; 1−(A+B+C) = 0.012; A:B.C = 0.204; A:C.B = 0.006; B:C.A = <0.001; A:B:C = 0.707.

Partial regression analyses showed that, compared to PSN and TEPMDE, MTWM totally and independently explained much more of the elevational variation of the percentage of temperate genera (87.7% and 9.6% vs 86.3% and 0.7%, and vs 26.5% and 3.2%, respectively) (Fig. 7).

Partial regression analyses also showed that, compared to percentages generated by MDE (represented by TEPMDE), climatic factors (represented by MTWM and PSN) totally and independently explained much more of the elevation variation observed in the percentage of plant genera with temperate affinity (93% and 69.7% vs 26.5% and 3.2%, respectively) (Fig. 8).

## DISCUSSION

The present study showed that with the increase of elevation, the percentages of plant genera with tropical affinity decreased, while those of plant genera with temperate affinity increased. Other relevant studies also demonstrated similar trends in the percentages of plant genera with tropical and temperate affinities in East Himalayan regions,

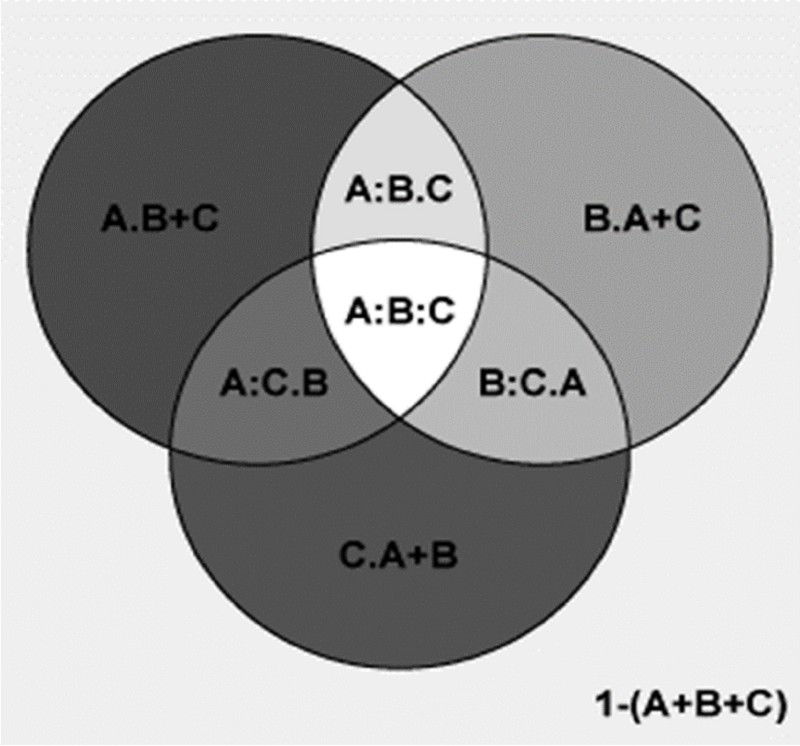

**Figure 7 Comparing the explanatory power of MTWM, PSN, and TEPMDE on the percentages of plant genera with temperate affinity by partial regression with the predictors included in the best model.** A = 0.877; 1−(A) = 0.12; A.B = 0.067; A:B = 0.813; A.C = 0.69; A:C = 0.189; A.B+C = 0.096; A:B+C = 0.784; B = 0.863; 1−(B) = 0.137; B.A = 0.05; B:A = 0.813; B.C = 0.601; B:C = 0.262; B.A+C = 0.007; B:A+C = 0.856; C = 0.265; 1−(C) = 0.735; C.A = 0.076; C:A = 0.189; C.B = 0.003; C:B = 0.262; C.A+B = 0.032; C:A+B = 0.233; A+B = 0.93; 1−(A+B) = 0.07; A+B.C = 0.697; A+B:C = 0.233; A+C = 0.955; 1−(A+C) = 0.045; A+C.B = 0.099; A+C:B = 0.856; B+C = 0.866; 1−(B+C) = 0.134; B+C.A = 0.082; B+C:A = 0.784; A+B+C = 0.962; 1−(A+B+C) = 0.038; A:B.C = 0.595; A:C.B = −0.029; B:C.A = 0.043; A:B:C = 0.218.

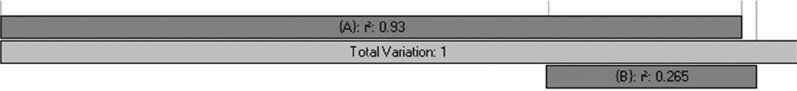

**Figure 8 Comparing the explanatory power of climatic factors (MTWM, MAP) and the percentages generated by MDE (TEPMDE) on the percentages of plant genera with temperate affinity by partial regression with the predictors included in the best model.** A shows the effects of climatic factors; B shows the effects of MDE. Total variance explained by {A} = 93%; Total variance explained by {B} = 26.5%; Total variance explained by {A+B} = 96.2%. [A.B] variance explained by {A} only = 69.7%; [A:B] Variance Sharely explained = 23.3%; [B.A] Variance explained by {B} only = 3.2%; [1−(A+B)] Unexplained variance = 3.8%.

for example, the Gaoligong Mountains (*Wang, Tang & Fang, 2007*), Yao Mountains (*Feng & Xu, 2008a*), and Nanghun River Nature Reserve (*Feng & Xu, 2008b*). In addition, no study conducted so far observed different elevational patterns in the percentages of plant genera with tropical and temperate affinities from the trends observed here. Therefore, the contrasting linear trends of the percentages of plant genera with tropical

and temperate affinities on elevation might be a universal rule, especially in the Himalayan regions, though further studies are needed in the future to corroborate this notion.

Although a variety of climatic factors, including energy, water, and seasonal factors, strongly affected the elevational patterns of the percentage of plant genera with tropical affinity, energy variables, especially MTCM, played the strongest roles representing winter coldness. These effects may be linked to the eco-physiological traits inherited from the biogeographical affinities of the tropical plants. Most plant genera with tropical affinity originated from and evolved in tropical regions, and prefer a warm climate or habitat (*Latham & Ricklefs, 1993a*, *1993b*; *Wiens & Donoghue, 2004*; *Wang et al., 2011*). Therefore, with increasing elevation, plant genera with tropical affinity may be gradually filtered out of the biomes by increasing freezing effects. It may suggest that the elevational patterns of the percentages of plant genera with tropical affinity and factors driving them supported the revision of the freezing-tolerance hypothesis.

Interactions between plant genera with tropical and temperate affinities may also play important roles in shaping the contrasting patterns of the percentages of plant genera with tropical and temperate affinities along elevation. With increasing elevation, increasing freezing may result in the decrease of habitat suitability for plant genera with tropical affinity, while plant genera with temperate affinity, which have stronger freezing tolerability and weaker warmth tolerability than plant genera with tropical affinity, gain competitive ability for habitat against plant genera with tropical affinity. We thus observed decreasing and increasing trends in the percentage of plant genera with tropical and temperate affinities with elevation, respectively, in which climatic gradients, especially the gradients of the factors representing freezing and warmth effects, regulate the interactions between plant genera with tropical and temperate affinities. Thus, we suggested that the research of the elevational patterns of the percentages of plant genera with tropical and temperate affinities should consider the roles of both abiotic filtering effects and biotic interactions between plant genera with tropical and temperate affinities.

Water availability played the strongest role in the hump-shaped diversity patterns of plant genera with tropical affinity in the similar elevational gradient in Nepal (*Li & Feng, 2015*). In contrast, in the present study, biotic competition and winter freezing may have played important roles in the decreasing percentages of plant genera with tropical affinity along the elevational gradient. Also, in Nepal energy availability in quadratic models strongly shaped the hump-shaped patterns of genus richness of plant genera with temperate affinity along similar elevational gradient (*Li & Feng, 2015*). Nonetheless, in the present study, all energy variables in their linear forms played strong roles in the increasing percentages of temperate plant genera. This may imply that energy variables play different roles in shaping elevational patterns of the percentages and richness of plant genera with temperate affinities. The quadratic terms of energy variables for the diversity of plant genera with temperate affinity may reflect the eco-physiological traits of plant genera with temperate affinity and their adaptation to the energy gradients on elevation. The linear terms of energy variables (especially MTWM) for the percentages of plant genera with temperate affinity may imply the elevational variation of competition between plant genera with tropical and temperate affinities. Overall, the elevational

patterns of tropical and temperate diversities and percentages may also be controlled by different mechanisms.

Previous studies suggested that plant genera with tropical affinity could be observed in high altitude, and plant genera with temperate affinity could occur at low altitude (*Wang, Tang & Fang, 2007*; *Li & Feng, 2015*; *Feng et al., 2016*), although their proportions might be lower than that at low and high altitude, respectively. Thus, although plant genera with temperate affinity may survive warm climates and plant genera with tropical affinity may survive cool or cold climate, they have lower competitive ability than plant genera with tropical and temperate affinities in such habitats, respectively. Therefore, we suggest that the percentages of plant genera with tropical affinity at high altitude and plant genera with temperate affinity at low altitude might be more influenced by the biotic interactions between them than the abiotic filtering effects.

Due to the increasing overlapping of ranges of organisms toward the center of the gradients, the MDE predicted hump-shaped altitudinal biodiversity patterns in the absence of other factors (*Colwell & Lees, 2000*; *Colwell, Rahbek & Gotelli, 2004*). In addition, the increasing overlapping toward the center of the gradients might be strengthened by wider ranges of organisms (*Colwell, Rahbek & Gotelli, 2004*; *Wang, Tang & Fang, 2007*). A previous study in Nepal found that when compared with plant genera with tropical affinity, plant genera with temperate affinity had wider elevational ranges (*Li & Feng, 2015*). This may imply that the plant genera with temperate affinity examined in the present study might be more strongly affected by MDE than plant genera with tropical affinity, especially around the midpoint of the elevational gradient. Therefore, we observed higher richness of plant genera with temperate affinity around the midpoint of the elevational gradient, which was reflected in the hump-shaped elevational pattern of the percentages of plant genera with temperate affinities, while the percentage of plant genera with tropical affinity evidenced a depression-shaped pattern. Our analyses showed that neither hump-shaped elevational pattern of TEPMDE nor the depressed-shaped elevational pattern of TRPMDE showed strong influences on the linear patterns of the percentages of plant genera with tropical or temperate affinities along an elevational gradient in Nepal. In addition, they explained less of the elevational variation of tropical and temperate plant genera percentages than climatic factors, particularly regarding the linear decreasing energy factors on elevation, which might play important roles not only in freezing-tolerance for plant genera with tropical affinity, but also in regulating biotic interactions between plant genera with tropical and temperate affinities. Therefore, freezing-tolerance and the interactions between plant genera with tropical and temperate affinities regulated by climatic factors played stronger roles than MDE in the elevational patterns of tropical and temperate plant genera percentages in Nepal.

## CONCLUSIONS

In the present study, we investigated the influences of climatic factors on the elevational patterns of the percentages of plant genera with tropical and temperate affinities in Nepal, and tested the revised version of the freezing-tolerance hypothesis for plant genera with tropical affinities. We found that the decreasing trends of the percentages of tropical

plant genera with increasing elevation were mainly controlled by MTCM, supporting the revised version of the freezing-tolerance hypothesis. The elevational patterns of diversities and percentages of plant genera with tropical and temperate affinities are likely controlled by different factors or mechanisms. Climatic factors played stronger roles than MDE in the elevational patterns of tropical and temperate plant genera percentages in Nepal.

## ACKNOWLEDGEMENTS

We would like to thank Fengshu Zha, Zhao Zhang, and Renyong Nan for their time and expertise during the preparation of this manuscript.

### Funding

This study was supported by the National Natural Scientific Foundation of China (Grant numbers 31560178 and 31360143 to Jianmeng Feng). The funders had no role in study design, data collection and analysis, decision to publish, or preparation of the manuscript.

### Grant Disclosure

The following grant information was disclosed by the authors:
National Natural Scientific Foundation of China: 31560178 and 31360143 to Jianmeng Feng.

### Competing Interests

The authors declare that they have no competing interests.

### Author Contributions

- Yunyun Lai conceived and designed the experiments, performed the experiments, analyzed the data, contributed reagents/materials/analysis tools, prepared figures and/or tables.
- Jianmeng Feng conceived and designed the experiments, performed the experiments, analyzed the data, contributed reagents/materials/analysis tools, authored or reviewed drafts of the paper, approved the final draft.

### Data Availability

   Climate data is from WorldClim (www.worldclim.org).
   Plant data is from the online version of the Annotated Checklist of the Flowering Plants of Nepal (http://efloras.org/flora_page.aspx?flora_id=110).

### Supplemental Information

Supplemental information for this article can be found online at http://dx.doi.org/10.7717/peerj.6116#supplemental-information.

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
