# Peer review of "Elevational patterns of the percentages of plant genera with tropical and temperate affinities in Nepal"

_PeerJ, doi:10.7717/peerj.6116_

## Round 0.1 · original submission · Major Revisions

Your manuscript has been carefully reviewed by three experts in this area. The last two reviewers are particularly critical but constructive in their reviews. Please pay attention to the comments from all three reviewers and carefully revise your manuscript. Note that if not revised satisfactorily, the manuscript can still be possibly rejected.

Reviewer 1 ·

Basic reporting

no comment

Experimental design

no comment

Validity of the findings

no comment

Additional comments

In this study, the authors related the percentages of tropical and temperate genera in 100-m elevational bands to several climate variables. They found that the decreasing trends of the percentages of tropical plants with increasing elevation were mainly controlled by mean temperature of the coldest month, and the increasing trends of the percentages of temperate plants with increasing elevation were mainly controlled by mean temperature of the warmest month. It appears worth documenting these patterns in the literature. English in the manuscript needs to be improved. Below are a few specific comments.

Specific comments
Abstract: More details about methods should be mentioned.
Line 91: change “known” to “know”
Line 170: delete one from “one one”
Lines 230 and 237: delete “A”
Line 251: delete though from “though though”

Reviewer 2 ·

Basic reporting

I am regret that I can not recommend this MS for publication, at least in it’s present version. See below.

Experimental design

Not very good.

Validity of the findings

Major weakness in methods.

Additional comments

This manuscript describe the percentages of tropical and temperate plants along an elevational gradient in relation to climatic indices. I am regret that I can not recommend this MS for publication, at least in it’s present version.

The topic itself is interesting. However, the MS did not provide insightful analyses and discussions that can improve our understanding of the altitudinal pattern. The whole MS is mainly talking about two correlations (though some other climate indices were used), and the conclusions are simply based on the correlation analysis in Table 1. The MS reads not very interesting.

There are some clear weakness in methods:
1) The MS conducted only correlation analysis without any multivariate models. However, in Methods and Results, the MS mentioned that they used VIF to diagnose collinearity. VIF is used to avoid collinearity among variables in multivariate models, but there’s no multivariate model in the MS at all.
2) The data used by the MS, the percentage, is well known to suffer to some major statistic problems. Results based on correlation analysis thus may not be reliable, as demonstrated by previous analyses (e.g. some publications by Gaston). The authors may consider some methods based on null model for a more robust analysis for this kind of data.
3) The data for percentages of tropical and temperate plants along elevational gradient is obtained through interpolation of plant occurrence. Consequently, the data will inevitably be affected by the interpolation effect (in some papers named as the mid-domain effect). This effect has strong influence on elevational richness pattern, and can strongly alter the correlations of richness with environmental factors. The percentages used in this MS is calculated from species numbers. Consequently, if the influence of the interpolation effect can not be well addressed, it is not reliable to draw conclusions on the major climate drivers of the percentages based on some simple correlations.
Consequently, I am sorry that I can not recommend this MS to be published in it’s present version.

Reviewer 3 ·

Basic reporting

Using the elevational distributions of over 1300 seed plant genera, the authors estimated the elevational patterns in the percentages of tropical and temperate genera, and explored the determinants of these patterns. The authors tested two different hypotheses respectively explaining the percentages of temperate and tropical genera. The issue addressed in this manuscript is interesting. I have a few comments for the consideration of the authors to improve the manuscript.

Experimental design

As far as I understand, the classification system of biogeographical affinities of seed plant genera developed by Wu is only for Chinese plants, not for global plants. To identify the biogeographical affinities of the genera that are not included in the classification system of Wu, the author used another system developed by Wielgorskaya. Are these two different systems comparable in the definition of biogeographical affinities?

The number of species within genera are different. More importantly, the species numbers of each genus within different elevational belts are also different. The authors only used the number and proportion of genera within each elevational belt. Could the proportions of genera really represent the proportions of different affinities in each elevational belt? If the proportions of species with different affinities were used, would the findings remain the same? Moreover, will the findings reported remain consistent for species with different lifeforms, e.g. woody vs. herbaceous species and annuals vs. perennials?

The authors proposed a new hypothesis named “warming-tolerance hypothesis”, which suggest that species with temperate affinities may could not tolerant the warm climate in low elevations. However, transplanting studies suggest that many species at high elevations could survive the warm climate at lower elevation or at more southern latitudes when they are planted in monoculture in botanical gardens. This observation makes many authors believe that the absence of temperate species from low elevations or latitudes is because temperate species living in warming regions have lower competitive ability than warm-adapted species. I think the authors could try to compare these two alternative hypothesis in explaining the elevational patterns in the percentages of temperate species.

Validity of the findings

no comment

Additional comments

According to the section of “statistical methods”, it seems that the authors conducted multiple regression analysis to select the best model explaining the percentages of tropical and temperate species. however, none of the results on multiple regressions were included in the manuscript or the supplementary materials. I suggest that these results should be included.

The English writing of this manuscript should be improved before publication.

Lines 81 – 84, these two sentences repeated some sentences in the first paragraph. Is it possible to merge the first and second paragraphs into one.
Line 106, insert “along elevational gradients” before and
Line 110, this sentence is not very clear. Consider to rephrase it.
Lines 129, 149, change “around” to “at”
Line 173, change “number” to “numbers”
Line 174, remove the first “and”.
Line 202, change resulting to resulted.
Line 251, remove the first “though”
Line 276, change comma to period.
Line 292, changes “elevational diversity” to “elevational diversity gradient”
Line 297, remove “terms of the”
Figure 1, Why are there white areas in the lowest part of the study area?

---

## Round 0.2 · Major Revisions

The revised manuscript has been reviewered by two experts. Please pay special attention to the comments from reviewer 2, which I agree. Reviewer 3 pointed out the writing should be much improved. Note that if not revised satisfactorily, the manuscript will be rejected without further review.

Reviewer 2 ·

Basic reporting

see below

Experimental design

see below

Validity of the findings

see below

Additional comments

The manuscript has been improved. But there are still some weakness in data analyses and writing, before it is accepted.

1) The MS used the MDE model and tried to resolve the problem of interpolation effect and the statistic problem associated with percentage data. If this method is acceptable (which is still controversial), the MS did not clarified their method to calculate the percentage of the tropical and temperate species.
When using this method, we can use the MDE model to predict the richness for tropical (MDEtrop) and temperate genera (MDEtemp) separately, and for the two group together (MDEtotal). Then we have two choice:
(1) Percentage of tropical genera = MDEtrop/ MDEtotal
(2) Percentage of tropical genera = MDEtrop/ (MDEtrop+MDEtemp)
Since the range size frequency distribution of tropical, temperate genera and the two group together should be different, it is possible that the two methods will get different results. If the results are different, which one should the MS use?
The MS did not clarified about this, and only said that “We simulated genus richness over the elevational gradient for all tropical and temperate genera separately.”

2) The MS has made some multivariate analyses in this version, and they made conclusions based on these multivariate analyses. But these results were listed supporting information. This is surprising.

3) The MS proposed the warming-tolerance hypothesis. Is this really a good hypothesis? The cold tolerance hypothesis has sound biological basis, and reflects niche conservatism. However, I am not convinced that fewer temperate species in warmer region was mainly caused by filtering by warmth. Many temperate species can live in gardens (without competition) that are much warmer than their natural distribution regions.

Reviewer 3 ·

Basic reporting

The authors have addressed most my concerns in the revised manuscript. I think the manuscript has been substantially improved. However, the writing is still not good enough for final publication. Many sentences are not well written. i have listed some below.

Experimental design

no comment

Validity of the findings

no comment

Additional comments

Line 24, Abstract: change “Geographical patterns of organisms” to “Geographical patterns of species diversity”
Lines 26-28, "but the factors or underlying mechanisms controlling the elevational patterns of the percentages of plant genera with tropical and temperate biogeographical affinities have attracted little attention”, could be reworded as “but the factors driving elevational patterns of the percentages of plants with tropical and temperate biogeographical affinities have not been adequately explored”
Line 34, Abstract: “tropical and temperate plant genera” should be “plant genera with tropical and temperate affinities”
Line 35, “percentage of tropical plants” should be “percentage of plant genera with tropical affinity”. Genera with tropical affinity is not the same as tropical genera, because a genus with tropical affinity may contain species that occupy temperate regions. I suggest the authors check this writing throughout the manuscript.
Line 38, Similarly, “temperate plants” should be “plant genera with temperate affinity”
Line 41, “tropical and temperate percentages” is not clear. Consider to reword it.
Line 44, Change “associated controlling factors” to “the factors driving them”.
Lines 43-Abstract, the first sentence of the Discussion section is awkward. Consider to split it into two sentences?
Line 90, “at mid-elevation” should be “along elevational gradients”; line 91, gradient should be gradients.
Line 105, change “controlling factor” to “driver of it”
Line 108, “hump-shaped patterns of elevational diversity” should be “hump-shaped elevational patterns”
Line 112, in their percentage in regional floras
Lines 154 – 169, the consistency between the two data sources for biogeographical affinities should be clearly stated in the text so that readers could judge the reliability of the results.
Line 230, remove “represented by climatic factors”
Lines 225 – 236, because partial regressions analyses were actually not conducted at all, these sentences starting from “Using the predictors ……” should be removed.
Lines 237 – 240, this sentence is not clear. Consider to split it into two as the relationship between the two clauses is hard to understand.
Lines 296 – 297, change it to “no study conducted so far observed different elevational patterns in the percentages of tropical and temperate plants from the trends observed here”
Lines 320 – 325, these sentences repeat previous ones. I suggest that they should either be removed, or integrated with the previous paragraph.
Lines 337 – 349, these sentences also repeats the introduction and the logic between them is hard to follow. I think these sentences could be greatly shortened.

---

## Round 0.3 · accepted · Accept

Congratulations. Thanks for your revision. It is good for publication now.

# Reviewer 3 ·

Basic reporting

no comments

Experimental design

no comments

Validity of the findings

no comments